# Aqueous Extract of Freshwater Clam Increases Alcohol Metabolism in Rats in a Preclinical Model

**DOI:** 10.3390/nu17111915

**Published:** 2025-06-03

**Authors:** Pei-Ying Chung, I-Chen Chiu, Ching-Yi Kuan, Tsung-Meng Wu, Kuo-Chan Tseng, Shu-Ting Chuang, Sen-Wei Tsai, Yu-Kuo Chen

**Affiliations:** 1Department of Obstetrics and Gynecology, Taichung Tzu Chi Hospital, Buddhist Tzu-Chi Medical Foundation, Taichung 427003, Taiwan; fishergirl88@gmail.com; 2Department of Food Science, National Pingtung University of Science and Technology, Pingtung 91201, Taiwan; chiu20010302@gmail.com (I.-C.C.); jlike5802@yahoo.com.tw (C.-Y.K.); tkc@mail.npust.edu.tw (K.-C.T.); 3Department of Aquaculture, National Pingtung University of Science and Technology, Pingtung 91201, Taiwan; wzm@mail.npust.edu.tw; 4Department of Nursing, Taichung Tzu Chi Hospital, Buddhist Tzu-Chi Medical Foundation, Taichung 427003, Taiwan; 5Department of Nursing, Tzu Chi University, Hualien 970374, Taiwan; 6School of Medicine, Tzu Chi University, Hualien 970374, Taiwan; tsaisenwei@gmail.com; 7Department of Physical Medicine and Rehabilitation, Taichung Tzu Chi Hospital, Buddhist Tzu-Chi Medical Foundation, Taichung 427003, Taiwan; 8Department of Post-Acute Care Center, Taichung Tzu Chi Hospital, Buddhist Tzu-Chi Medical Foundation, Taichung 427003, Taiwan

**Keywords:** alcohol metabolism, freshwater clam, alcohol dehydrogenase, acetaldehyde dehydrogenase, health risks

## Abstract

Excessive drinking or even alcoholism poses noticeable health risks to society, and investigating ways to improve alcohol metabolism may be a potential strategy to address this public health issue. The aim of this study was to explore the efficacy of freshwater clam aqueous extract (CE) in promoting alcohol metabolism and to further elucidate its potential mechanism. Male Wistar rats were divided into four groups: (1) control group (C); (2) vehicle group (V), which was given a single dose of 2 g/kg bw ethanol; (3) low-dose CE group (CEL), which was given ethanol and 128 mg/kg bw CE; and (4) high-dose CE group (CEH), which was given ethanol and 256 mg/kg bw CE. Blood was drawn from the tails of the rats at 0, 1, 2, 3, and 4 h after alcohol administration, and serum samples were collected. The results showed that compared with the V group, oral administration of CE reduced the ethanol concentration in the serum of the rats, with the area under the serum ethanol curve (AUC) of the CEH group decreased by 32.6%, exhibiting a significant difference (*p* < 0.05). Moreover, the high-dose CE (CEH) treatment significantly increased the activities of alcohol dehydrogenase (ADH), acetaldehyde dehydrogenase (ALDH), catalase (CAT), and superoxide dismutase (SOD) in the liver of the rats by 41.5%, 42.4%, 40.6%, and 34.6%, respectively, compared with those in group V (*p* < 0.05). The Western blot results indicated that CE reduced the expression of ethanol-induced inflammation-related proteins COX-2, iNOS, and TNF-α in the liver by 66.4%, 90.6%, and 41.4%, respectively. In conclusion, it can be inferred that CE can help reduce the ethanol concentration in the serum of rats fed with alcohol, and its possible mechanism is to promote the metabolism of ethanol by increasing the activities of ADH and ALDH and the antioxidant capacity in the liver.

## 1. Introduction

Excessive drinking can lead to many adverse physical symptoms, such as headaches, dizziness, and lightheadedness [1,2,3]. In severe cases, it may even endanger the life and safety of oneself and others [4]. When the amount of alcohol intake exceeds the body’s metabolic capacity, the alcohol concentration in the blood rises, and the alcohol enters the brain through the blood circulation, resulting in drunkenness [5]. Beyond this, high levels of alcohol in the body disrupt cellular function and the antioxidant system. Alcohol metabolism in the body produces inflammatory mediators and cytokines, which may induce liver cell injury [6]. Over the years, several animal models have been established by researchers to investigate diverse diseases caused by alcohol [7]. The most common pathophysiological approach performed to study the influences of alcohol and its mechanisms is to administer alcohol to animals, either acutely or chronically, since both long-term (i.e., chronic) and one-time (i.e., acute) alcohol consumption can affect the functions of organs in our body [8]. Among them, acute animal models can be used to explore the effects of natural products or functional supplements on short-term drinking, such as binge drinking.

Once alcohol is absorbed by the digestive system into the body, it is transported to the liver through the blood circulation and oxidized into a toxic metabolite, acetaldehyde, mainly through alcohol dehydrogenase (ADH) in liver cells [9]. Acetaldehyde is then metabolized by acetaldehyde dehydrogenase (ALDH) to relatively non-toxic acetate [10], which is finally converted to CO_2_ and water and exhaled through the respiratory system or excreted in the urine [11]. Although metabolites produced during alcohol metabolism may affect hangovers and cause liver dysfunction, natural products or functional foods that can initiate or promote alcohol metabolism should be able to alleviate hangovers and ultimately reduce liver damage [12]. Under certain conditions, cytochrome P450 2E1 (CYP2E1) is involved in the metabolism of alcohol. During the metabolic process, reactive oxygen species (ROS) are generated, which reduces the GSH content and increases lipid peroxidation in the liver [13]. Therefore, long-term drinking, which not only stimulates the production of ROS but also disrupts the homeostasis of the redox system, is one of the main causes of oxidative stress in the body [14]. If substances can promote alcohol metabolism, inhibit inflammatory substances, and enhance antioxidant enzyme activity, they could potentially ameliorate the consequences of drunkenness and even alcohol-related diseases.

Freshwater clams (*Corbicula fluminea*), also called golden clams and Asiatic clams, are extensively grown in the muddy bottoms of rivers, ponds, and lakes in freshwater or brackish water ecosystems (close to river mouths) of Asian countries [15]. They are suitable for artificial breeding due to their fast growth and high reproductive capacity. However, freshwater clams are quite sensitive to water quality, so farms with clear and clean water for cultivation are required. As several studies have shown that freshwater clams contain high levels of nutrients and regulate blood lipids, protect the liver, inhibit cancer growth, and improve gastric ulcers [16,17,18,19], they have been developed into various health foods in the current market. Chen et al. indicated that freshwater clam aqueous extract (CE) contains amino acids (or peptides), phenolic compounds, and phytosterols [20], which are considered to show antioxidant activity and might have the potential to alleviate the redox system imbalance induced by alcohol [21]. Furthermore, it was also recorded in the Compendium of Materia Medica (Bencao Gangmu in Chinese) that clams have medicinal effects on the metabolism of alcohol-induced toxicity. In the present study, we used CE as experimental material to explore its effects on blood ethanol content in rats fed with a single dose of alcohol and to further elucidate its potential mechanism.

## 2. Materials and Methods

### 2.1. Chemicals and Reagents

An Ethanol Assay Kit was provided by Abnova (Taipei, Taiwan). An Alcohol Dehydrogenase Activity Colorimetric Assay Kit and an Aldehyde Dehydrogenase Activity Colorimetric Assay Kit were obtained from BioVision (Milpitas, CA, USA). Superoxide dismutase (SOD) and catalase (CAT) assay kits were provided by Cayman Chemical (Ann Arbor, MI, USA). The COX-2 antibody was obtained from Cell Signaling Technology (Danvers, MA, USA). The iNOS and TNF-α antibodies were purchased from Santa Cruz Biotechnology (Santa Cruz, CA, USA). The β-actin antibody, RIPA lysis buffer, ammonium persulfate (NH_4_)_2_S_2_O_8_), and secondary antibody were purchased from Merck Millipore (Burlington, MA, USA). The GE Healthcare AmershamTM ECL prime Western blotting detection reagent, Cytiva Amersham™ Hybond™ PVDF membrane, and Coomassie assay protein reagent were obtained from Thermo Fisher Scientific (Waltham, MA, USA). Sodium dodecyl sulfate (SDS), ethanol, methanol, and all other chemicals and reagents were purchased from Sigma-Aldrich (St. Louis, MO, USA).

### 2.2. Experimental Material Preparation

The freshwater clam aqueous extract (CE) was a gift from Li Chuan Aquafarm Co., Ltd. (Hualien, Taiwan), which was prepared following the procedure previously published by Isnain et al. [19]. Briefly, ddH_2_O was used to extract dried freshwater clam meat powder at a ratio of 1:20 (*w*/*v*) for 24 h, and then the filtrate was obtained after centrifugation and filtration. The filtrate was dried with a lyophilizer after freezing to obtain CE, which was then kept at −20 °C and orally administered to rats after being resuspended in ddH_2_O.

### 2.3. Animals and Treatment

Five-week-old Wistar male rats were purchased from BioLASCO Co., Ltd. (Taipei, Taiwan) and housed in the animal room of National Pingtung University of Science and Technology. The environment was controlled under a 12 h light/dark cycle, room temperature was maintained at 22 ± 2 °C, and relative humidity was 60–80%. The animals were allowed to access food and water freely. After the rats were stably maintained for 1 week, they were divided into four groups (6 rats in each group): (1) in the control group (C), rats were given ddH_2_O; (2) in the vehicle group (V), rats were given a single dose of ethanol (2 g/kg bw) and ddH_2_O; (3) in the low-dose CE group (CEL), rats were given ethanol and 128 mg/kg bw CE; (4) in the high-dose CE group (CEH), rats were given ethanol and 256 mg/kg bw CE. On the day of the experiment, the vehicle (ddH_2_O) and CE were administered to rats (fasted overnight) through gavage 20 min after the ethanol treatment. Serum was collected from the tail artery at 0, 1, 2, 3, and 4 h after ethanol treatment. At the end of the experiment, all animals were sacrificed and their blood and liver tissues were collected. 

All animal studies were conducted under the animal use protocol (NPUST-105-024) approved by the IUCUC of National Pingtung University of Science and Technology and adhered to the ARRIVE guidelines. For each animal, four different investigators were involved as follows: the first investigator administered the treatment based on the randomization table. This investigator was the only person aware of the treatment group allocation. A second investigator was responsible for the alcohol and CE treatment procedure, while a third investigator performed the blood collection procedure. Finally, a fourth investigator conducted the subsequent analysis.

### 2.4. Measurement of Serum Ethanol Concentration

The ethanol concentration of the serum samples from rats was analyzed using the Ethanol Assay Kit. In brief, 90 μL of the working reagent (containing 80 μL assay buffer, 1 μL enzyme A, 1 μL enzyme B, 2.5 μL NAD and 14 μL MTT) was added to 10 μL of the serum sample, and the mixture was allowed to react at room temperature for 30 min after mixing gently and thoroughly. Then, 100 μL of the stop reagent was used to terminate the reaction. The absorbance was read at a wavelength of 565 nm using an ELISA reader, and the ethanol concentration in the serum was calculated using the formula provided by the manufacturer (Appendix A).

### 2.5. Analysis of Hepatic Alcohol Dehydrogenase (ADH) and Acetaldehyde Dehydrogenase (ALDH) Activities

The ADH and ALDH activities in the liver tissue of the rats were analyzed using the ADH and ALDH Activity Colorimetric Assay Kits according to the manufacturer’s instruction manual. Hepatic tissue (50 mg) was added to 200 μL of ice-cold assay buffer and homogenized, followed by centrifugation (13,000× *g*, 10 min) to remove insoluble matter. Then, 2 μL of liver homogenate was transferred to the 96-well plate, and 100 μL of reaction mixture (containing 82 μL ADH assay buffer, 8 μL developer, and 10 μL substrate) was added to each well and allowed to react at 37 °C. The absorbance was measured at a wavelength of 450 nm using an ELISA reader, and the activities of ADH and ALDH were calculated using the formula provided by the manufacturer (Appendix A).

### 2.6. Analysis of Hepatic Catalase (CAT) and Superoxide Dismutase (SOD) Activities

For the analysis of gastric CAT activity, 10 mg of liver tissue was added to 100 μL of buffer (containing 1 mM EDTA and 50 mM K_2_HPO_4_) and homogenized according to the procedure in our previous study [22] with slight modifications. After centrifugation at 10,000× *g* for 15 min at 4 °C, the supernatant for CAT activity analysis was obtained. To prepare the homogenate of gastric tissue for SOD activity determination, 30 mg of liver tissue was transferred into the homogenizing tube filled with beads, then 270 μL of buffer (containing 12 mM K_2_HPO_4_, 8 mM KH_2_PO_4_, and 1.5% KCl) was added and subsequently homogenized. The supernatant was then obtained with centrifugation of the homogenate for 30 min at 16,400× *g* at 4 °C. Then, the CAT and SOD activities of the supernatant fraction were analyzed using a commercial assay kit according to the manufacturer’s instructions (Appendix A).

### 2.7. Analysis of Hepatic Protein Expression

After the liver tissue was homogenized with lysis buffer, the protein level of the homogenate was determined following the procedure previously reported by Tsai et al. [23]. The above protein solution was mixed with 1/6 volume of sample buffer, heated at 100 °C for 10 min, cooled in an ice bath, and then subjected to SDS-PAGE. Then, the proteins were transferred from the gel to a polyvinylidene difluoride (PVDF) membrane at 300 mA for 2 h. After transfer, the PVDF membrane was blocked with 5% skim milk powder solution [soaked in PBST (phosphate-buffered saline with Tween 20)] at room temperature for 1 h. Then, the primary antibody (anti-β-actin, anti-COX-2, anti-iNOS, or anti-TNF-α) diluted with skim milk powder solution was added to react at room temperature overnight. After reacting with the primary antibody, the PVDF membrane was washed with PBST followed by incubation in a secondary antibody solution for 1–2 h. Thereafter, the membrane was incubated with ECL reagent for 1 min, and the signal of the target proteins was captured with a luminescent imaging system (Hansor, Taichung, Taiwan). ImageJ^®^ software (version 1.44, National Institute of Health, Bethesda, MD, USA) was applied to quantify the signal intensity of the target proteins, and the relative expression of proteins was normalized according to the loading control bands of β-actin.

### 2.8. Statistical Analysis

The data were subjected to one-way analysis of variance (ANOVA) and Duncan’s multiple comparison test to check for significant differences among treatments (*p* < 0.05). Data are expressed as means ± standard deviations (SDs).

## 3. Results

### 3.1. Effect of CE Supplement on Serum Ethanol Concentration in Rats

Figure 1A shows the effect of oral administration of CE on the ethanol concentration in rat serum. Blood samples were collected at 0, 1, 2, 3, and 4 h after treatment with 2 g/kg bw of ethanol, and the serum ethanol concentration was determined. The results showed that the ethanol treatment remarkably increased the ethanol concentration in the serum of the rats, which reached its peak at 1 h after ethanol-loading, then gradually decreased and reached the lowest level after 4 h. The serum ethanol concentration of the rats in the vehicle group (V) was as high as 0.55 ± 0.04 g/L in the first hour, while the administration of CE reduced the serum ethanol levels in the rats. The serum ethanol concentrations of the rats in the onefold CE (CEL) and twofold CE (CEH) groups were 0.50 ± 0.03 g/L and 0.39 ± 0.04 g/L, respectively. There was a significant difference between the CEH group and the V group (*p* < 0.05), and this trend was maintained until the second and third hours, but there were no significant differences among all groups after the fourth hour (*p* > 0.05). The area under the serum ethanol curve (AUC) represents the extent of total exposure to ethanol in the rats over a given period of time. The results shown in Figure 1B indicate that the AUC of the rats in the CEH group is significantly lower than that in the V group (*p* < 0.05), while there is no significant difference in AUC between the CEL group and the V group.

### 3.2. Effect of CE Supplement on Hepatic ADH and ALDH Activities in Rats

One of the main pathways of alcohol metabolism in the human body is that when blood flows through the liver, ALDH catalyzes the metabolism of alcohol into acetaldehyde [24]. In the present study, we measured the ADH activity in rat liver to clarify whether CE can reduce the serum ethanol concentration in rats by activating the activity of enzymes related to alcohol metabolism. As shown in Figure 2A, the hepatic ADH activity of ethanol-fed rats in the V group was slightly increased compared with that of the rats in the C group. After the CE administration, the activity of hepatic ADH in the rats was further increased, and a significant difference was found between the CEH group and the V group (*p* < 0.05). It is suggested that administration of CE increased the ADH activity in the liver, thereby accelerating the metabolism of ethanol and, thus, its clearance in the serum of the rats. Moreover, we further analyzed the ALDH activity in the liver and found that in the rats in the CEH group, it was significantly higher than in the V group (Figure 2B; *p* < 0.05). ALDH is a critical enzyme for the body to convert the more harmful acetaldehyde, a major factor involved in a series of unpleasant effects, into the relatively non-toxic acetate [25].

### 3.3. Effect of CE Supplement on the Activities of Hepatic CAT and SOD in Rats

The effects of CE on the CAT and SOD activities in the liver of rats are shown in Figure 3. Alcohol loading remarkably reduced the activities of CAT and SOD in the liver (group C vs. group V). Administration of CE at either low or high doses (group CEL or group CEH) significantly increased the activities of CAT and SOD (*p* < 0.05). After alcohol consumption, ethanol flows through the blood to the liver, and a large amount of ethanol is converted into acetaldehyde, which accumulates in the liver and promotes the generation of reactive oxygen species (ROS), leading to oxidative stress and liver damage [26]. CAT and SOD are known to be the major antioxidant enzymes in the liver that can protect against diverse oxidants yielded during alcohol metabolism [27]. Furthermore, oxidative stress caused by alcohol has been demonstrated to deplete antioxidant enzymes in the body, including CAT and SOD [28], which were decreased by alcohol loading in this study. Administration of CE significantly restored the activities of hepatic CAT and SOD, implying that CE alleviated alcohol-induced damage of the antioxidant enzyme system in the liver of rats.

### 3.4. Effect of CE Supplement on Expression of TNF-α, COX-2, and iNOS Proteins in Liver of Rats

Figure 4A shows the effect of CE administration at 256 mg/kg bw on hepatic TNF-α protein expression in alcohol-fed rats. The expression of TNF-α in group V was remarkably higher than that in group C, and CE administration significantly reduced the alcohol-induced increase in TNF-α protein expression (*p* < 0.05). We also found that alcohol enhanced the expression of COX-2 and iNOS in the liver of the rats (group V vs. group C; Figure 4B,C). Several studies have revealed that alcohol treatment increased the expression of the inflammatory markers COX-2 and iNOS in liver cells [29,30,31,32], suggesting that elevated production of these two markers is involved in alcohol metabolism. CE administration significantly decreased the expression of COX-2 and iNOS in the liver of the rats in this study, implying that CE may help to reduce the inflammatory status associated with alcohol metabolism.

## 4. Discussion

In the present study, we administered a single dose of alcohol (2 g/kg bw) to rats as an acute preclinical model to investigate whether an aqueous extract of freshwater clam (CE) could accelerate the metabolism of alcohol. Our results showed that the ethanol treatment remarkably increased the ethanol concentration in the serum of rats, and the administration of CE reduced the serum ethanol levels and AUC in these rats compared with those in group V (Figure 1). In addition to reducing excessive alcohol use, relieving symptoms associated with drinking is also a useful strategy. Despite the anatomical, physiological, and genetic differences between animals and humans that make research results irreproducible [33], animal experiments are the biological model most often used to explore the possible effects of short-term and long-term alcohol consumption on our bodies. How to reduce the burden from harmful use of alcohol has been included in one of the critical indicators in Sustainable Development Goals (SDGs) health target 3.5 [34]. A number of studies have been conducted to develop or investigate substances that accelerate the metabolism of alcohol in the body and thereby reduce the burden of alcohol [16,35,36]. An increase in blood alcohol concentration is a physiological phenomenon following alcohol consumption. 

Since ancient times, freshwater clams have been considered to possess hepatoprotective effects in Asia, and even the Compendium of Materia Medica (Bencao Gangmu in Chinese) noted that freshwater clams have medicinal properties in alcohol detoxification. The safety of long-term use of freshwater clam extract has also been confirmed by researchers [37]. Chijimatsu et al. [38] also showed that freshwater clam extract accelerated the clearance of ethanol in the blood of rats acutely treated with ethanol. Moreover, freshwater clam protein hydrolysate was reported as a potential functional food for enhancing alcohol metabolism in an acute alcohol exposure animal model [39]. 

Freshwater clams have been widely documented to contain several biologically active ingredients, such as phytosterols, polysaccharides, carotenoids, essential amino acids, and peptides [18,40,41], which may be one of the main factors that enable freshwater clams to promote alcohol metabolism. Our previous study also pointed out that CE has a certain amount of taurine [19], which has been found to stimulate the metabolism of alcohol. 

Both ADH and ALDH are key catalytic enzymes in the reaction process of metabolizing alcohol into non-toxic acetate [10]. In the present study, we found that the administration of CE enhanced the activities of hepatic ADH and ALDH in rats, especially in the CEH group (Figure 2), which showed significant differences compared with group V (*p* < 0.05). Any natural substances or products that can increase the activity of ADH and ALDH in the liver should have a positive effect on alcohol metabolism and reduce the concentration of alcohol in the blood [42]. Red ginger, Korean pear, mango, asparagus, and fenugreek have been found to increase the activities or expression of ADH and ALDH, thereby alleviating hangovers and their associated symptoms [43,44,45,46,47]. It was also noted in a previous study that a protein hydrolysate prepared from the meat of freshwater clams could be a potential natural product for improving alcohol metabolism through the enhancement of hepatic ADH and ALDH activities [39]. The results in the present study showed that CE administration through gavage remarkably enhanced the activities of ADH and ALDH in the liver of rats, implying that CE could be a candidate nutraceutical for accelerating alcohol metabolism in the body or relieving hangovers. 

During alcohol metabolism, ROS are produced in the body, which disrupt the homeostasis of the redox system [14]. Our results indicated that alcohol treatment reduced the activities of CAT and SOD in the liver of rats (Figure 3), while the administration of low- or high-dose CE significantly restored the activities of CAT and SOD (*p* < 0.05). Bai et al. [48] found that pretreatment with proteoglycan, a polysaccharide isolated from *C. fluminea*, significantly increased the activities of SOD and CAT as compared with the alcohol administration group. It is suggested that CE and its potential active components may help to improve alcohol-induced redox system homeostasis imbalance through restoring CAT and SOD activities. In addition to producing ROS and reducing antioxidant enzymes, consuming alcohol also activates pro-inflammatory cytokine TNF-α and the enzymes COX-2 and iNOS [49]. Figure 4 shows that the protein expression of TNF-α, COX-2, and iNOS was up-regulated by a single dose of alcohol (2 g/kg bw), and CE administration significantly down-regulated their expression in the liver of rats (*p* < 0.05). The fatty acids in CE were found to have an inhibitory effect on the NF-κB pathway and iNOS and COX-2 expression, thereby suppressing the development of inflammation [50]. Our previous study also showed that indomethacin-induced overexpression of COX-2, iNOS, and TNF-α proteins in gastric mucosa was reduced by the administration of CE [19]. Our findings indicate that CE plays a crucial role in the protection against pro-inflammatory responses during alcohol metabolism.

## 5. Conclusions

The results of this study showed that the administration of CE could reduce the serum alcohol concentration in rats fed alcohol. The possible mechanism may be related to the fact that CE can increase the activities of ADH and ALDH in the liver of rats, thereby accelerating the metabolism of alcohol. Moreover, CE was found to increase the activities of CAT and SOD in the liver and reduce the protein expression of COX-2, iNOS, and TNF-α, indicating that CE may help to improve the antioxidant status of rats and is beneficial to the metabolism of alcohol. The possible beneficial effects of CE need to be further investigated in chronic alcohol models or human clinical trials, including the effects of CE on energy metabolism and gut microbiota.

## Figures and Tables

**Figure 1 nutrients-17-01915-f001:**
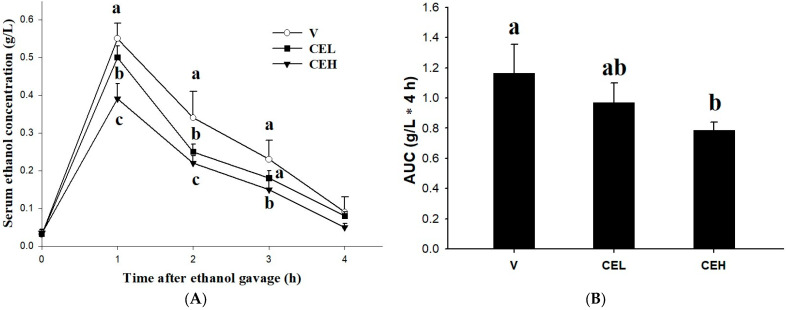
Effect of CE on (**A**) serum ethanol concentration and (**B**) area under the curve (AUC) in rats fed a single dose of ethanol (2 g/kg bw). In the vehicle group (V), rats were given a single dose of ethanol (2 g/kg bw) and ddH_2_O; in the low-dose CE group (CEL), rats were given ethanol and 128 mg/kg bw CE; in the high-dose CE group (CEH), rats were given ethanol and 256 mg/kg bw CE. Data are expressed as means ± SDs (n = 6). Symbols or bars with different letters indicate significant differences among groups (*p* < 0.05).

**Figure 2 nutrients-17-01915-f002:**
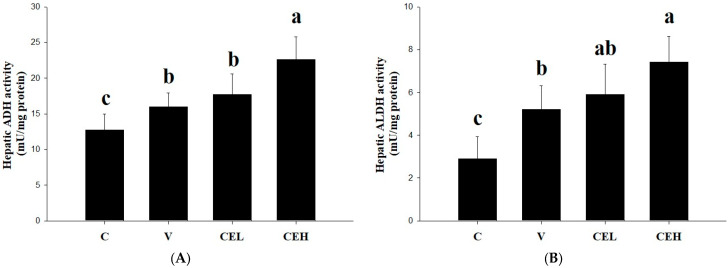
Effect of CE on the activities of (**A**) ADH and (**B**) ALDH in the liver of rats fed a single dose of ethanol (2 g/kg bw). In the control group (C), rats were given ddH_2_O; in the vehicle group (V), rats were given a single dose of ethanol (2 g/kg bw) and ddH_2_O; in the low-dose CE group (CEL), rats were given ethanol and 128 mg/kg bw CE; in the high-dose CE group (CEH), rats were given ethanol and 256 mg/kg bw CE. Data are expressed as means ± SDs (n = 6). Bars with different letters indicate significant differences among groups (*p* < 0.05).

**Figure 3 nutrients-17-01915-f003:**
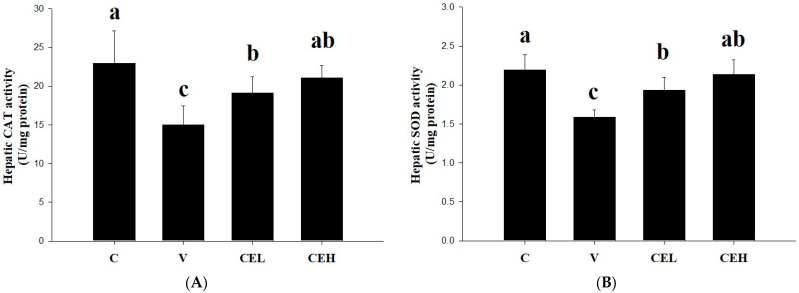
Effect of CE on the activities of (**A**) CAT and (**B**) SOD in the liver of rats fed a single dose of ethanol (2 g/kg bw). In the control group (C), rats were given ddH_2_O; in the vehicle group (V), rats were given a single dose of ethanol (2 g/kg bw) and ddH_2_O; in the low-dose CE group (CEL), rats were given ethanol and 128 mg/kg bw CE; in the high-dose CE group (CEH), rats were given ethanol and 256 mg/kg bw CE. Data are expressed as means ± SDs (n = 6). Bars with different letters indicate significant differences among groups (*p* < 0.05).

**Figure 4 nutrients-17-01915-f004:**
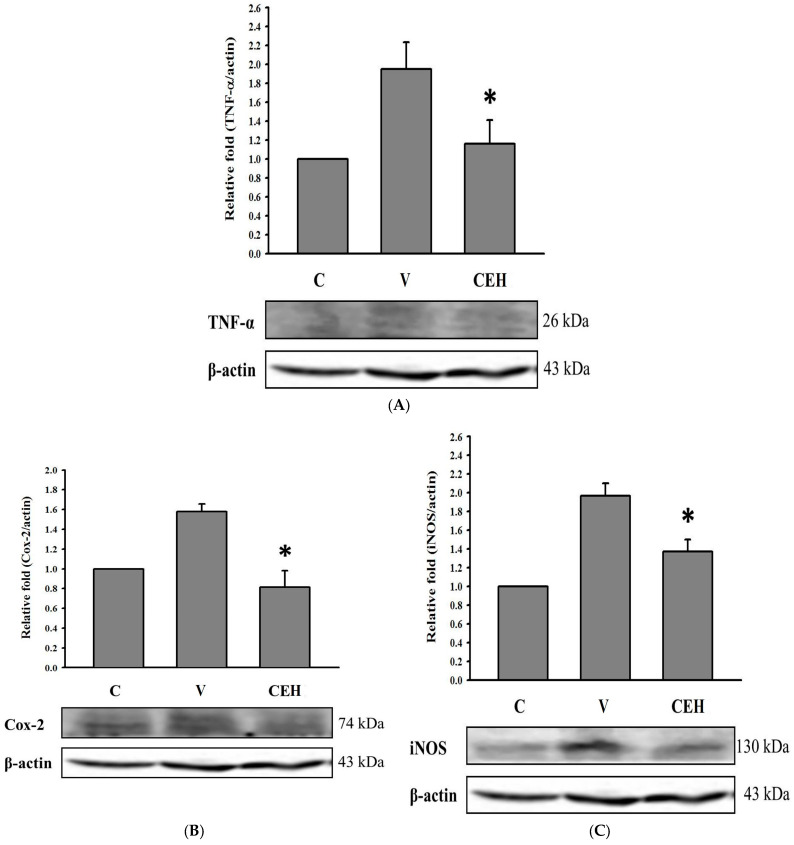
Effect of CE on the protein levels of (**A**) TNF-α, (**B**) COX-2, and (**C**) iNOS in the liver of rats fed a single dose of ethanol (2 g/kg bw). In the control group (C), rats were given ddH_2_O; in the vehicle group (V), rats were given a single dose of ethanol (2 g/kg bw) and ddH_2_O; in the high-dose CE group (CEH), rats were given ethanol and 256 mg/kg bw CE. Data are expressed as means ± SDs (n = 3). * *p* < 0.05 compared with group V.

## Data Availability

Data are contained within the article.

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
