# Peer review of "Aqueous Extract of Freshwater Clam Increases Alcohol Metabolism in Rats in a Preclinical Model"

_nutrients, 2025, doi:10.3390/nu17111915_

Round 1
Reviewer 1 Report
Comments and Suggestions for Authors
This is an interesting original article. The aim of the study was to explore the efficacy of freshwater clam aqueous extract (CE) in improving alcohol metabolism and to elucidate its mechanism. The results provide evidence that oral aministration of CE reduced the ethanol concentration in rats' serum. The putative mechanism is related to increase of activity of enzymyes responsible for alcohol metabolism. Furthermore, it was shown that CE treatment could reduce expression of ethanol-induced inflammation-related proteins in the liver. They concluded that CE treatment improves alcohol metabolism in the liver, reducing the risk of unwanted side effects.
The article is acceptable after a small revision. Some points need to be addressed:
1) The examined rats were given a single, high dose of alcohol. However, problems due to an excessive use of alcohol may be a consequence of long-term, everyday consumption, of even small amounts of alcohol. Therefore, it would be of interest to study a population of rats being systematically intoxicated by smaller amounts of alcohol. Do the Authors think that such a study is possible to perform ?
2) What is the meaning of the symbol "ab" in Figures 1B, 2B and 3(A,B) ?
3) What is the meaning of symbols: "C", "NC" and "CEM" in the Figure 4 ?
Minor point: it should probably be "CEL", not "CEM" group (Page 5, line 5 from the top)
Reviewer 2 Report
Comments and Suggestions for Authors
Chung et al. present a manuscript within the scope of this Journal. The manuscript presents an adequate description of an important preclinical model. However, the authors should improve the manuscript in several aspects:
-The title of the manuscript is too simplistic. The authors should focus the title according to the results obtained. The authors should include the word "preclinical model."
-The authors should improve the abstract of the manuscript. It should be appropriately written and should highlight the conclusions.
-The introduction is too simplistic and should focus on both the pathophysiological mechanisms and the need for this animal model.
-The authors should use updated references.
-The methodology is described in a very simple and poorly reproducible manner. It should be improved in numerous aspects.
-The animal model should be described in detail.
-The sample size should be calculated using statistical methods.
-The statistical potential should be calculated and described.
-The authors must describe the statistical methodology and justify it point by point.
-The authors must include all ELISA method protocols in the supplementary material.
-The quantification methods are not described for precise scientific reproducibility.
-The results must be adequately described, with high-quality figures.
-The authors must provide a detailed discussion in a separate section. Conclusions should be included in points 4 and 5.
-A graphic summary must be included.
-The authors must improve their use of English grammar in a precise and specialized manner.
Reviewer 3 Report
Comments and Suggestions for Authors
The study aimed to investigate the efficacy of freshwater clam aqueous extract (CE) in enhancing alcohol metabolism and to clarify its underlying mechanism. Several significant issues necessitate attention below:
Abstract
Page 1; Line 31-33: The sentence about "oral administration of CE reduced the ethanol concentration..." could better highlight the effect size or percentage reduction for clarity.
Page 1; Line 34-36: The increase in ADH, ALDH, CAT and SOD activities should be given either in percentage or statistical values.
Page 1; Line 37-38: The reduction in COX2, iNOS and TNF-α expression should be given either in percentage or statistical significance values.
Page 1; Line 39: Minor grammatical error: "the CE can effectively..." remove "the".
Introduction
Some parts of the background are repetitive (alcohol metabolism pathways are explained multiple times). (Page 2; Line 61, 68, 70 and Page 5; Line 206).
Page 2; Line 71: It would strengthen the introduction to briefly mention why a clam extract would theoretically promote alcohol metabolism (i.e., known active components).
Methods
Page 3; Line 100: Typographical errors ("Experimetal" should be corrected to "Experimental Material Preparation").
Page 3; Line 114: I suppose animal numbers are small (n=6 per group), especially considering variability in alcohol metabolism. Please acknowledge this limitation.
Page 3; Line 136: The sentence shouldn’t start with the numerical value ’50 mg’.
Page 4; Lines 143 and 153: The assay procedures in sections 2.6 and 2.7 were developed by the authors or used earlier published methods (required citations).
No mention of blinding procedures for researchers performing assays (this should be clarified for objectivity).
The western blot method section could describe loading control normalization more explicitly.
Ethical approval mentioned — good, but you should also mention adherence to ARRIVE guidelines.
Results
Figures could be better labelled. Figures 1–4 need to have fully descriptive captions ("Effect of CE on serum ethanol" is too brief — explain sample sizes, significance indicators, etc.).
Data presentation is clear, but exact p-values (where significant) would improve transparency ("p<0.05" is sometimes too general).
Consider reorganizing the “Results and Discussion” into separate sections for better clarity (unless journal format requires combination).
Discussion
Please expand on the potential limitations of translating findings from a rat model to humans.
No discussion of potential active compounds (e.g., taurine, peptides) beyond speculative comments — a brief paragraph would strengthen the scientific argument.
Long-term safety of clam extract use is not discussed.
Conclusion
You could strengthen the conclusion by briefly suggesting the next research step (e.g., chronic alcohol models, human trials).
Avoid overly strong language ("can effectively reduce”). It is better to use cautious wording ("may help to reduce").
References
Minor inconsistencies in formatting (spacing, punctuation) across references — careful proofreading needed.
Some duplicated citations (example: Isnain et al. 2022 is cited twice similarly — consolidate if appropriate).
Comments on the Quality of English LanguageThe quality of English language seems fine but require checking typoerror and correction
Round 2
Reviewer 2 Report
Comments and Suggestions for Authors
Accept in present form.
Author Response
Comments 1: Accept in present form.
Response 1: Thanks for the reviewers' comments which made this manuscript more complete.